# Fog Computing Capabilities for Big Data Provisioning: Visualization Scenario

Halimjon Khujamatov [1,2], Khaleel Ahmad [3], Nargiza Usmanova [1,2], Jamshid Khoshimov [4], Mai Alduailij [5,*] and Mona Alduailij [5]

1   Computer Engineering Faculty, Tashkent University of Information Technologies, Tashkent 100200, Uzbekistan; kh.khujamatov@gmail.com (H.K.); nargizausm@mail.ru (N.U.)
2   Telecommunication Technologies Faculty, Tashkent University of Information Technologies, Tashkent 100200, Uzbekistan
3   Department of Computer Science and Information Technology, Maulana Azad National Urdu University, Hyderabad 500032, India; khaleelahmad@manuu.edu.in
4   Ministry for Development of Information Technologies and Communications, Tashkent 100000, Uzbekistan; j.khoshimov@mitc.uz
5   Department of Computer Sciences, College of Computer and Information Sciences, Princess Nourah Bint Abdulrahman University, Riyadh 84428, Saudi Arabia; maalduailej@pnu.edu.sa
*   Correspondence: maalduailij@pnu.edu.sa

**Abstract:** With the development of Internet technologies, huge amounts of data are collected from various sources, and used 'anytime, anywhere' to enrich and change the life of the whole of society, attract ways to do business, and better perceive people's lives. Those datasets, called 'big data', need to be processed, stored, or retrieved, and special tools were developed to analyze this big data. At the same time, the ever-increasing development of the Internet of Things (IoT) requires IoT devices to be mobile, with adequate data processing performance. The new fog computing paradigm makes computing resources more accessible, and provides a flexible environment that will be widely used in next-generation networks, vehicles, etc., demonstrating enhanced capabilities and optimizing resources. This paper is devoted to analyzing fog computing capabilities for big data provisioning, while considering this technology's different architectural and functional aspects. The analysis includes exploring the protocols suitable for fog computing by implementing an experimental fog computing network and assessing its capabilities for providing big data, originating from both a real-time stream and batch data, with appropriate visualization of big data processing.

**Keywords:** fog computing; big data; visualization; cloud computing; performance; capability

## 1. Introduction

The rapid development of Internet technologies, and the ability to retrieve information anytime, anywhere, have changed the life of society. The Internet as hypermedia, with its web services and applications, is creating enormous amounts of data (big data) as a response to the tremendous changes in enabling technologies and encompassing advances. Gartner (https://www.gartner.com, accessed on 17 October 2021) defines big data as large volumes of various information assets, as well as the processing of such data that requires a cost-effective and innovative form. Obviously, collecting, processing, and storing these large and diverse sets of information, growing at ever-increasing rates, requires means other than traditional tools to support the 'three V's', i.e., volume, velocity, and variety of big data. Big data statistics worldwide [1] show that the rates of market coverage and penetration of impact on different businesses increased in the period 2010–2020; specifically, the volume of data interactions increased by 5000%. If we use the statistics of 2020, one Internet user generated 1.7 megabytes of data every second, and this gives a rough total of 2.5 quintillion bytes of data generated in one day. More than 90% of organizations are investing in big data and artificial intelligence; by 2023, big data market is predicted to be

worth approximately USD103 billion, and, in subsequent years, data volumes will reach more than 180 zettabytes.

The concept of big data involves more than just the analysis of huge amounts of information: data is created by and from a variety of sources, stored in many different repositories, and is mainly presented in a format that does not fit well with the traditional, structured database format. As a result, it becomes crucial to have the necessary tools to establish relationships between these data, and take meaningful outcomes for users. In addition, data are now being often updated, and traditional methods of information analysis cannot keep up with the huge amount of constantly updated data. In essence, the concept of big data implies working with a huge volume of information of diverse composition, very often updated and located in different sources, in order to increase work efficiency, create new products, and increase competitiveness.

The IT industry has undergone major changes over the past decade. Among the most influential was the emergence of cloud technologies for data storage, computing, and networking. However, the realities of the Internet of Things imply the necessity to evolve the new concept of fog computing, which processes data on network peripherals (such as personal computers, smartphones, personal gadgets, and so on) out of the cloud. The development of the IoT requires the support of the mobility of IoT devices for various locations, with small latency for data processing [2,3]. A new platform proposed to meet such requirements features data processing in the immediate vicinity of the sources, without the need to transfer them to large data centers. The evolving pace of growing the amount of data on today's Internet brings to the arena the necessity of providing scalable access to the computing resources, in order to process those data appropriately, on demand, anytime, anywhere, with the required performance.

The computing paradigm, so-called cloud computing, makes it possible to implement a 'pay-per-use' model over the Internet to provide computing resources on demand from consumers. Cloud systems accelerate innovation, and save a variety of physical and hardware computing resources, due to their scalability [4,5].

Due to the nature of data today, being extremely widely distributed, and delivered continuously, in large volumes, and to a huge number of users of all kinds of devices, it is challenging to make the cloud model effective, due to the need to deliver content to the users over a geographically dispersed platform, rather than through a cloud physically located in one place. Thus, the idea of fog computing is to distribute data and make them actually closer to the user, which can eliminate network delays, and the limitations of data transmission.

Fog is not an alternative to the cloud. On the contrary, fog has a fruitful relationship with the cloud, especially in administration and data analytics, and such interactions are spawning a new class of applications. The fog computing (FC) architecture can be seen as a layer on the border between the cloud and IoT devices. According to [6], fog computing is an extension of the cloud at its core, which has some specific computing resources and data storage registers, and is located at the nodes of networks. Currently, fog computing technologies are widely used in next-generation networks, vehicles, energy management, radio access networks, etc., and optimize the computational resources of different application systems [7,8]. FC also helps achieve sustainability through several application areas, while in the emerging IoT, it is the first choice for time-sensitive applications, due to it residing closer to the devices and sensors [9].

Fog computing is expanding the boundaries of cloud computing by making the network and data extremely widely dispersed by allowing a geographically dense distribution of computation and data. There are several reasons why the distributed fog computing infrastructure is useful: the possibility of analyzing big data faster and more efficiently; there is no need to create a complete network infrastructure for local users' needs; and it makes real-time data analysis more feasible, even when the infrastructure is actively scaling. In this context, the authors provide analysis of fog computing capabilities for big data provisioning, and describe the modeling scenario for visualizing the way to attract related

processes. The paper briefly outlines the related work in this research domain, emphasizing the potential and capabilities of FC for different problematic areas. The methodology is then proposed for exploring the layered architecture of FC, where the necessity to study the real-time nature of data processing is justified, and the big data processing framework developed for FC is outlined. Based on this, FC performance evaluation is performed, by demonstrating the capability of the fog node and its ability to process data. Result analysis is followed by presenting analysis of fog computing usages for big data processing scenarios, providing detailed introductions and experiments for FC capabilities in data analytics.

## 2. Related Work

Some applications require large amounts of network bandwidth, as well as very low latency, and naturally, these needs require data processing near the end terminals. Based on the concepts of fog computing, it is safe to say that fog computing implies a proximity to data generators. In most cases, such generators are IoT sensors. The term was initially coined by Cisco, and is seen as the concept of edge computing, wherein resources have placement between users and the cloud, in order to improve performance. In a general view, FC is similar to edge computing, with alterations made to where the intelligence and computing power is placed.

The potential and capabilities of FC within different problematic areas are considered and studied in different research papers and development projects. A recent report [10] outlines the framework for a comprehensive understanding of the edge computing ecosystem, with valuable insights on infrastructure, hardware, networking, and software. This report provides edge computing particulars from the experts in this field, including updates from the market and thoughts from industry leaders, a brief review of applications, and a glossary of terms related to edge computing.

The authors in [11] provide an overview of edge and fog computing concepts to address big data storage and analysis for smart cities, describing service delivery models that emphasize issues of latency and network bandwidth required by time-sensitive smart city applications. In [12], the authors discuss the background and motivations for fog computing, and present a reference architecture for FC by describing the services and related characteristics. Research directions and enablers in this research domain, such as programming models, security and reliability, resource management, and energy minimization, are outlined by shedding light on the commercial applications and case study scenario implementation.

The security and privacy of fog computing are considered in [13], wherein the authors study different types of attacks, and investigate the features of such attacks. It is shown in security scenarios that properly applied authentication techniques ensure the system is safeguarded. The authors in [14] consider fog computing model architecture while studying various applications in real-time scenarios, with the implied security challenges. Paper [15] outlines some of the main security and privacy challenges faced by fog and edge computing. The authors demonstrate security issues affecting the work and implementation of edge and fog computing.

Paper [16] presents the dimensions of big data, and how different needs and services are transformed using applications. It is demonstrated in the paper how cloud computing can be applied for the visualization of big data, and work with cloud computing capability. A methodology is presented to deal with appropriate functionality.

In addition, the aspects of modelling the infrastructure are also studied, e.g., [17–19]. The authors in [17] propose solutions to fog computing problems, including a blockchain-enabled federated learning model. It is assumed that global and local updates are stored on fog servers. Blockchain-related issues are discussed in [8] to provide persistency and audibility. In [18], the authors discuss security and privacy in an FC-enabled IoT environment. The reliability requirements are given in [19], including the implied latency when fog nodes work together. Furthermore, in [20], the analysis of fog computing at a higher

level, considering its use in cases and potential issues, is carried out, though the use of implementations. The authors in [21] conduct in-depth research studies with mathematical models for areas comprising low-latency devices; fog computing reduces latency and energy dissipation considerably, due to dwindled data transmission. Moreover, research studies of security issues outline several theoretical vulnerabilities.

Notwithstanding, little research into actual implementation has been performed. One of these implementations was completed by Cirani et al. [22], who created a fog node called 'IoT Hub'. The results derived from their experiment highlight low memory and processing requirements.

To better grasp the critical issues of FC capabilities, the actual implementation of fog computing, comprising a few fog nodes providing the big data gained from sensor networks and assessing the performance of protocols used for big data provisioning in fog computing, is within the scope of this research.

## 3. Methodology and Prerequisites for Modeling the FC Environment

For this research, a single board computer called Raspberry Pi was chosen as a fog node to create research conditions. The current implementation of fog computing uses several protocols, either standardized or experimental, to transmit real-time sensor data [23]. The following were potential candidates, namely, MQTT and CoAP. The first protocol suitable for FC is MQTT, designed for IoT in tandem machine-to-machine communications. MQ Telemetry Transport is a publish–subscribe messaging protocol, which is inherently used in networks that have the following characteristics: unreliability, high latency, small data, and data transfer nodes that are too remote. The minimum amount of service information, the availability of service classes, and the hierarchical structure of topics are the undeniable advantages of the MQTT protocol, which is confirmed by a wide variety of both client and server software, including open-source software. Its primary application is in sensor technologies that work with satellite technologies. While brokers receive information from the satellite, it is also used to automate home life, medicine, and much more. In addition, owing to its miniature size, low energy consumption, and a small number of data packets, along with the effective distribution of information to receivers, it is suggested as an excellent choice for mobile applications [24].

Following this, the experimental FC network is expected to follow the OpenFog Reference Architecture, which proposes some advantageous aspects compared to other counterparts. These benefits are combined into the term SCALE, standing for:

- Security (additional measures need to be taken to guarantee safe and trustworthy transactions);
- Cognition (the conscience of client-centric goals to enable autonomy);
- Agility (scalability of the overall infrastructure/pace of innovation);
- Latency (system control/real-time processing);
- Efficiency (dynamical free resource accumulation from participants).

Apart from the aforementioned criteria, the FC hierarchy is advised to be set up in accordance with one of the approaches introduced in [24]: Figure 1 shows the cloud-independent fog. This model is applicable in the cases of some events, such as when regulation, security, and confidentiality are required (military direction), and when the central cloud is unavailable in a certain territory. Figure 2 demonstrates a type of cloud that requires different time intervals in action.

Having chosen an appropriate model for the potential FC, a lambda architecture-based framework (http://lambda-architecture.net/, accessed on 1 February 2021) is proposed, with the purpose of processing big fata generated by various devices. At its core, lambda architecture involves processing big data using batch processing methods. This approach focuses on compensating for fault tolerance, throughput, and latency through packet processing. Broadly, the demand for lambda architecture is accelerated due to the growth of big data, real-time analytics, and motivation to dwindle the latencies of MapReduce. Due to the flexibility of lambda architecture (depicted in Figure 2), it can be adjusted to process

real-time big data streams; in other words, processing large data streams in real-time. From a high-level perspective, the infrastructure based on lambda architecture consists of three layers, as shown in Figure 2.

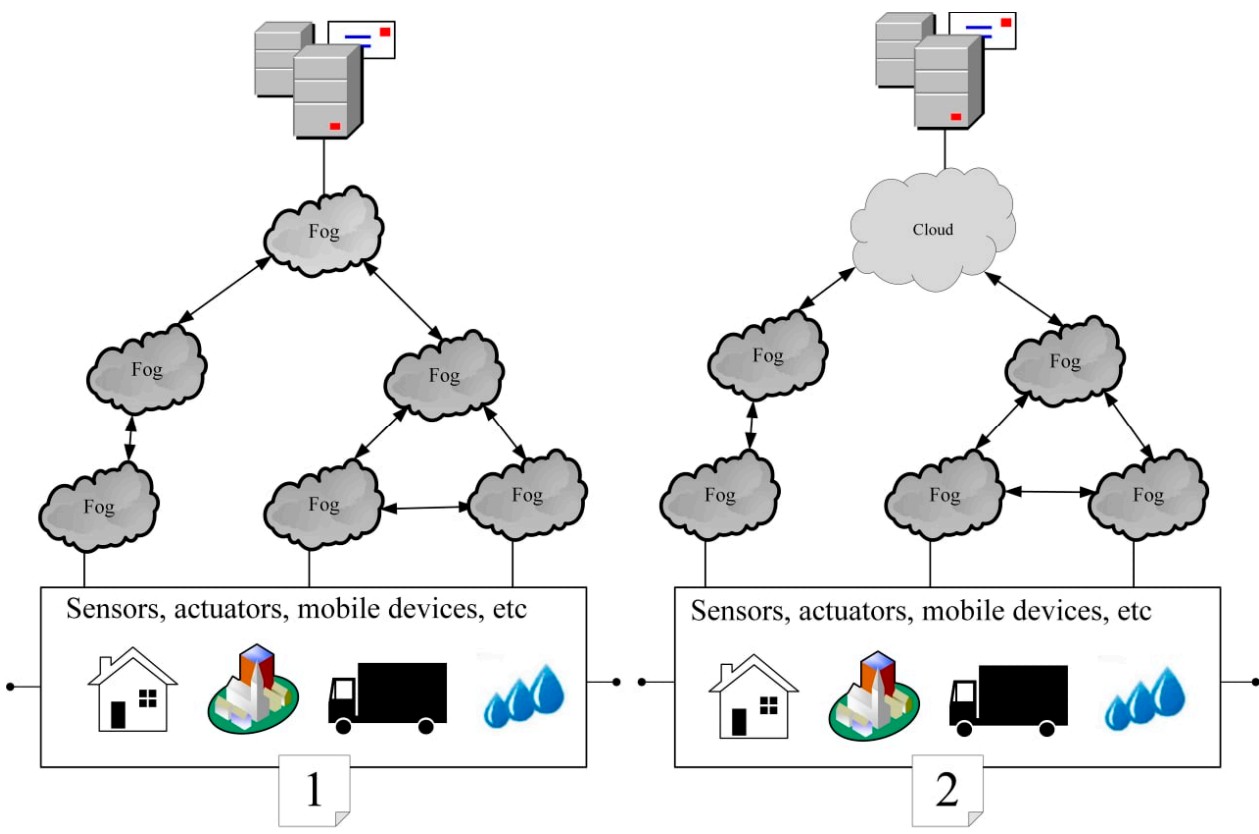

**Figure 1.** Examples of fog computing hierarchy.

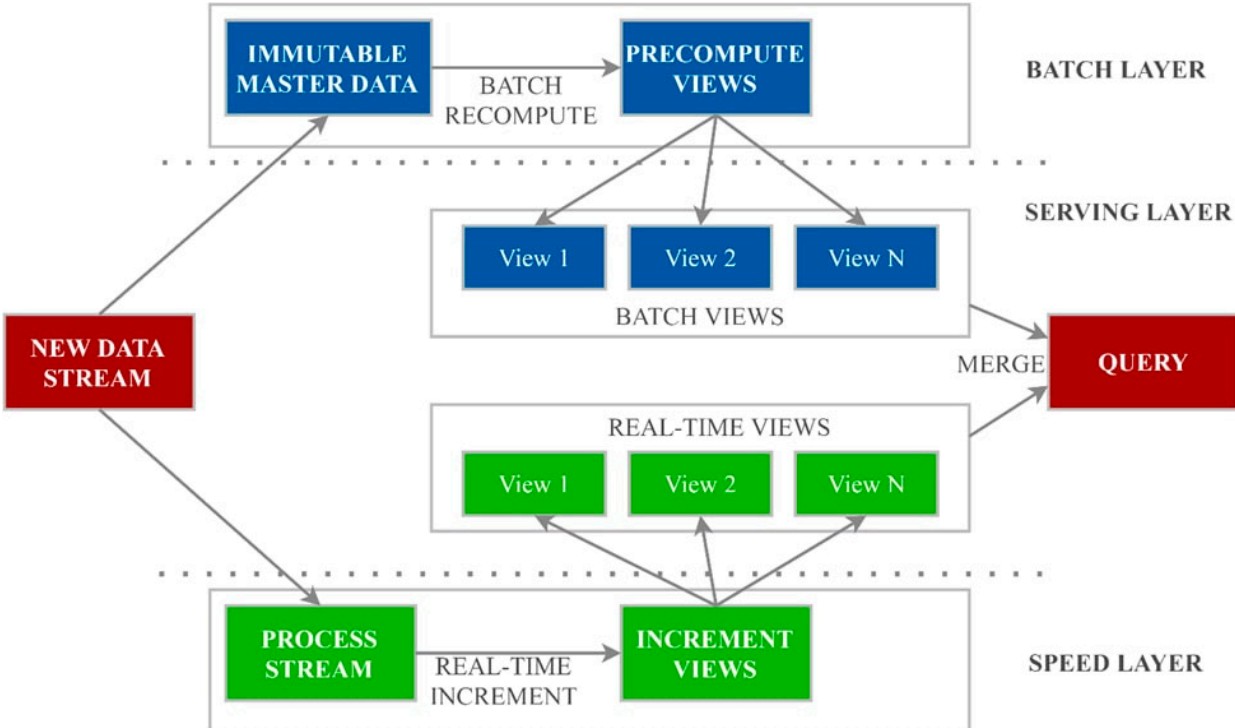

**Figure 2.** The structure of the lambda architecture designed for big data processing.

The batch layer is primarily responsible for storing immutable, increasing master datasets (HDFS), along with random computing views from the dataset (MapReduce). The calculation of the so-called views is a continuous operation; when new data arrive, the process of aggregation into views takes place. The knowledge gained from all datasets is computed at the batch level, and is not expected to frequently change the conclusions. In this case, the time spent on the calculation depends on the size of the processed data. The outcome of the previous layer is a collection of files comprising pre-processed knowledge. The second layer, called the serving layer, indexes and discloses the views, so that they can be queried. Batch views are static, meaning the maintainer provides batch updates as well as random reads. There are ideal tools for batch processing and maintenance layers, such as Hadoop and Impala. Hadoop is relevant because it can process and store petabytes of data, while Impala, in turn, interactively requests such data [25]. The real-time requirements for the batch and service layers are not met, since MapReduce itself has a significant delay, so hours of time can be required before the presentation data are propagated to the serving layer. It is for this reason that a layer with speed is needed. The last layer is the speed layer (Apache Storm, Spark Streaming), resembling the batch tier. The similarity is that either the batch or speed layer gains insight from the data received. The speed layer compensates for the shortcomings of the packet layer here, and the disadvantage is the delay at this level. The compensation is carried out due to the peculiarity of real-time streaming applications, such as Storm or others. The real-time insights include only the delta results to complement the batch views. One noteworthy point related to the term 'real-time' is its association with near real-time, delays in the occurrence of events, and the availability of processed data of the current event. The real-time mode in lambda architecture, by its very nature, enables the processing of large data, which are captured after the start of a batch iteration. The batch layer recalculates the batch views from scratch, while the speed layer uses an incremental model (in which the views increase depending on the received data). The feature of the speed level is characterized by the fact that real-time views are necessarily temporary, and after the data is propagated through the respective layers, the results in the views can be discarded here; all this implies real-time mode (Figure 3).

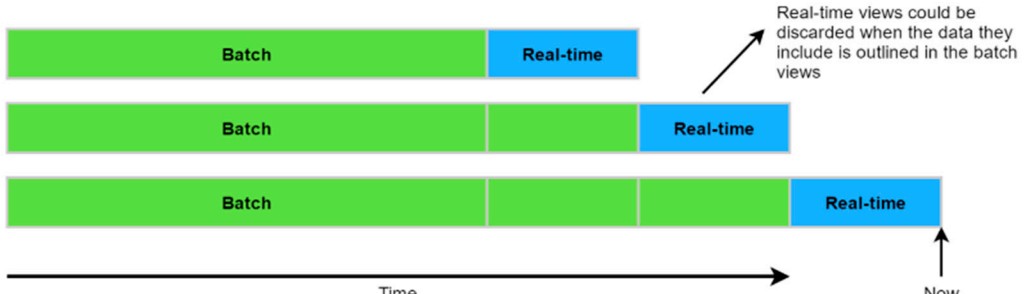

**Figure 3.** An illustration depicting the transient nature of real-time views.

The final piece of the puzzle is disclosing the real-time insights, so they can be queried and merged with the batch views to obtain in-depth knowledge. Since real-time views are incremental, the speed layer requires both random reads and writes, and for this purpose, Apache Kudu, Apache Cassandra, or Apache HBase are primarily preferred, which provide Apache Storm or Apache Spark streaming capability. For continuously growing real-time views, it can be requested to merge with batch views simultaneously. Meanwhile, the provisioning of raw data to be processed is performed using Apache Kafka and other data collecting frameworks, e.g., StreamSets Data Collector, allowing the routing and processing of data streams.

Considering the above-mentioned, a real-time big data processing framework for FC, which is provided in Figure 4, was developed. The figure represents the high-level structural model of real-time big data processing in a fog environment. The provision of big data is primarily made by StreamSets Data Collector, which supports several origins

i.e., Apache Kafka, MQTT, CoAP, HTTP, and UDP. While the batch processing layer is based on Apache Hadoop, the speed layer relies on either Apache Storm or Apache Spark streaming. Following this, Apache Cassandra and Apache Kudu real-time database management systems constitute the serving layer, which is then connected to the data visualization software supporting real-time data visualization and data preparation to present the data obtained from the lower layer of FC.

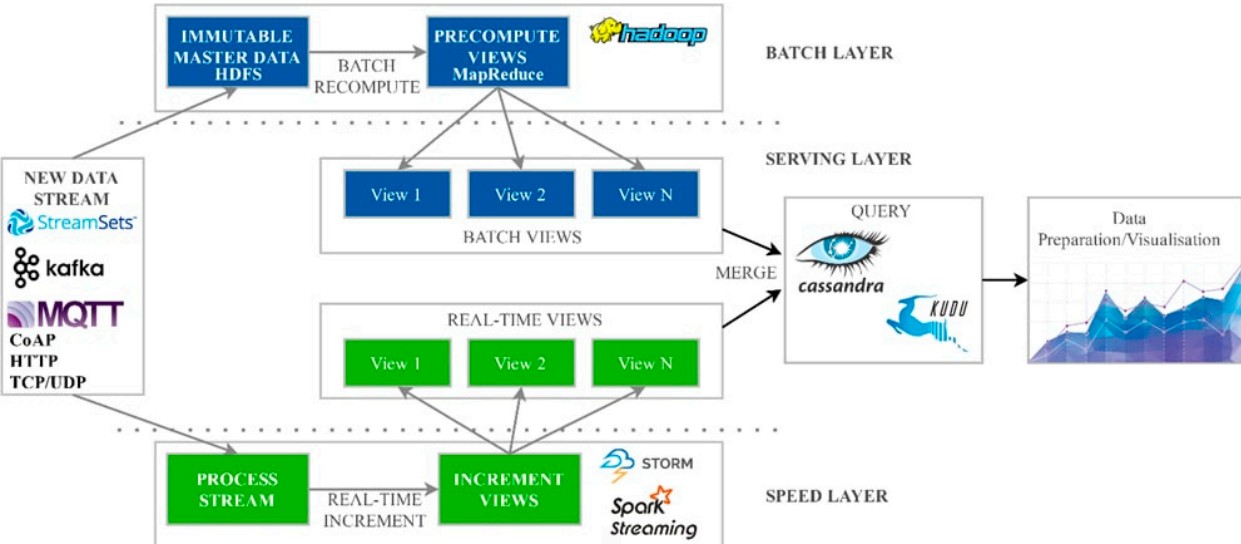

**Figure 4.** The real-time big data processing framework for the fog computing environment.

The estimation of the performance of FC is carried out on working fog nodes, and the incoming traffic is captured using a network analyzer, which is Wireshark v2.2.6 in this case. Having captured the network state in the high-loading, generated by sensors transmitting a vast amount of raw data through each of the protocols to be assessed, the capture file is analyzed on a desktop PC, using a language and software environment for statistical computing R, to reach a conclusion regarding FC performance in big data provisioning. In addition to the previous elements, the batch processing capabilities of FC were scrutinized by running a word count MapReduce operation and Pi estimation using the quasi-Monte Carlo method on a fog cluster. In this case, the time spent on the execution was measured in order to draw a reasonable conclusion on the basis of the result obtained from the simulation.

## 4. Fog Computing Performance Evaluation

In this subchapter, we create an experimental network based on the previous subchapter to evaluate the possibilities of fog computing. In this experimental fog, a cloud-independent model is chosen, which is shown in Figure 1. The parameters of this experimental network are as follows: StreamSets (data collector) fog node (the number of network elements is 100 threads) receives data at a speed of 100 Mbps, and the time allotted for the experiment is 450 s. The foggy cluster consists of six machines assigned to three nodes for data processing; the Raspberry Pi Model B is chosen. For the remaining nodes, Intel Core 2 Duo E8500 is chosen.

The following metrics are selected to measure the performance of big data provisioning in fog computing:

- throughput;
- the value of the transmitted information;
- jitter.

Here, the provision of raw data is significantly dependent on the protocols, since these data are directly dependent on these protocols. In this regard, the MQTT and CoAP

protocols of the Internet of things are comparable with the regular protocols HTTP and UDP. Accordingly, all these protocols are assigned the same data formats.

The simple raw data is data from temperature sensors that have the appropriate formats with JavaScript Object Notation (JSON) (Concise Binary Object Representation (CBOR) is also suitable in this case). All packets are analyzed using a real-time fog node packet analyzer, including a stream processing pipeline (Figure 5).

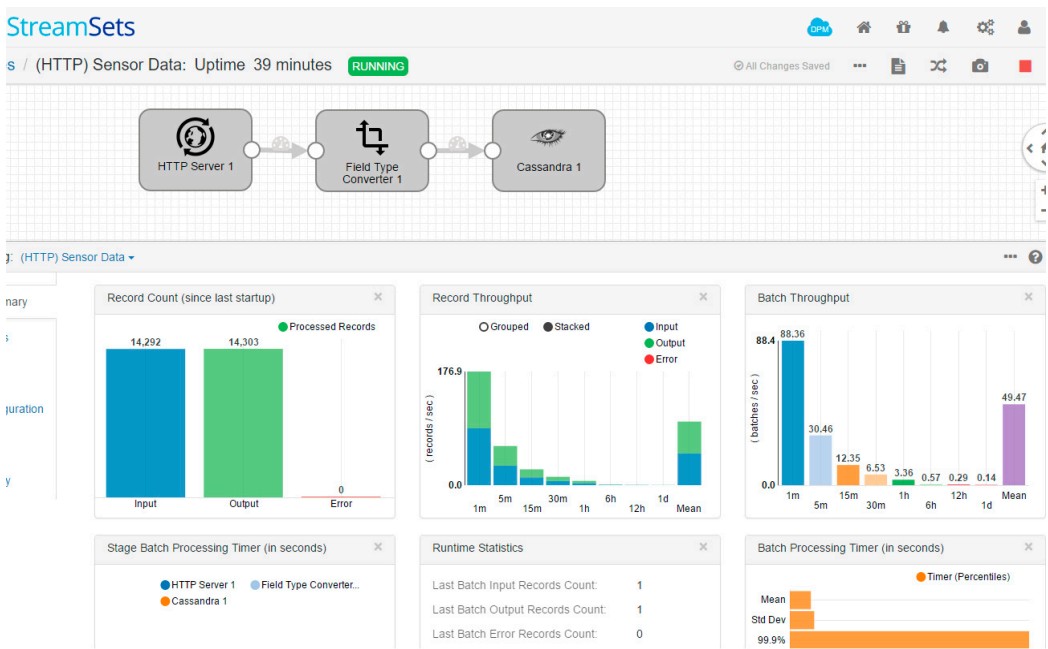

**Figure 5.** Data processing pipeline running on fog node.

Figure 5 shows the raw data being received by the HTTP server on the fog node, which is still processing streams, i.e., converting UNIX time to a convenient format. After time processing, real-time data is transferred to Apache Cassandra.

Data transfer to the database is omitted to eliminate the distortion of metrics, directing outgoing traffic to the basket, while all information is discarded. This approach prevents deviations in the experiment results, due to the lack of redundant data flow to be stored in the real-time data warehouse, which consumes a huge amount of network resources.

A further capability of the fog node is the ability of stream processing; to put it differently, the real-time data stream can be processed within strict time constraints. This allows for monitoring various processes in important areas of the government, such as banking. An example of the validation of credit card transactions performed by taxi services is provided in Figure 6. In this case, the transactions are obtained from the pseudo-real-time source generating credit card records on the fog node itself, to briefly demonstrate the existence of fog nodes' real-time stream processing capabilities. As shown in Figure 6, the pipeline classifies credit card details before their identification, in accordance with vendors. Following the classification, the credit cards with valid records are parsed to recognize the number of credit cards. It is then saved to the distributed file system, which is a Hadoop distributed file system (HDFS) in this implementation [25,26]. One point to be noted concerning this implementation is the feature of user notifications. The pipeline has several interactive charts depicting the number of records being processed, throughput, batch or stream processing time per record, and the total number of processed records (Figure 7).

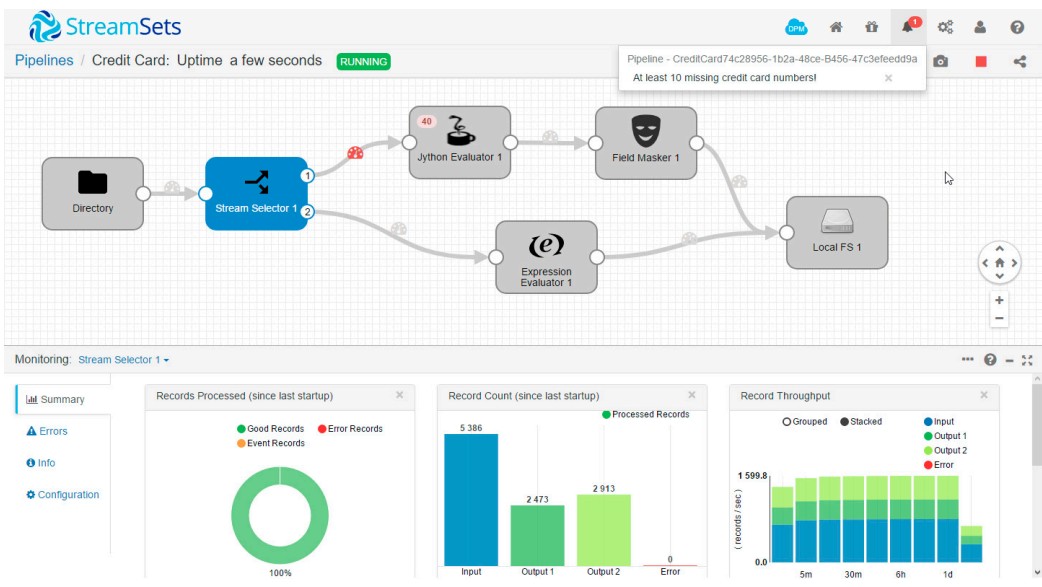

**Figure 6.** Stream processing pipeline of credit card validation running on the fog node.

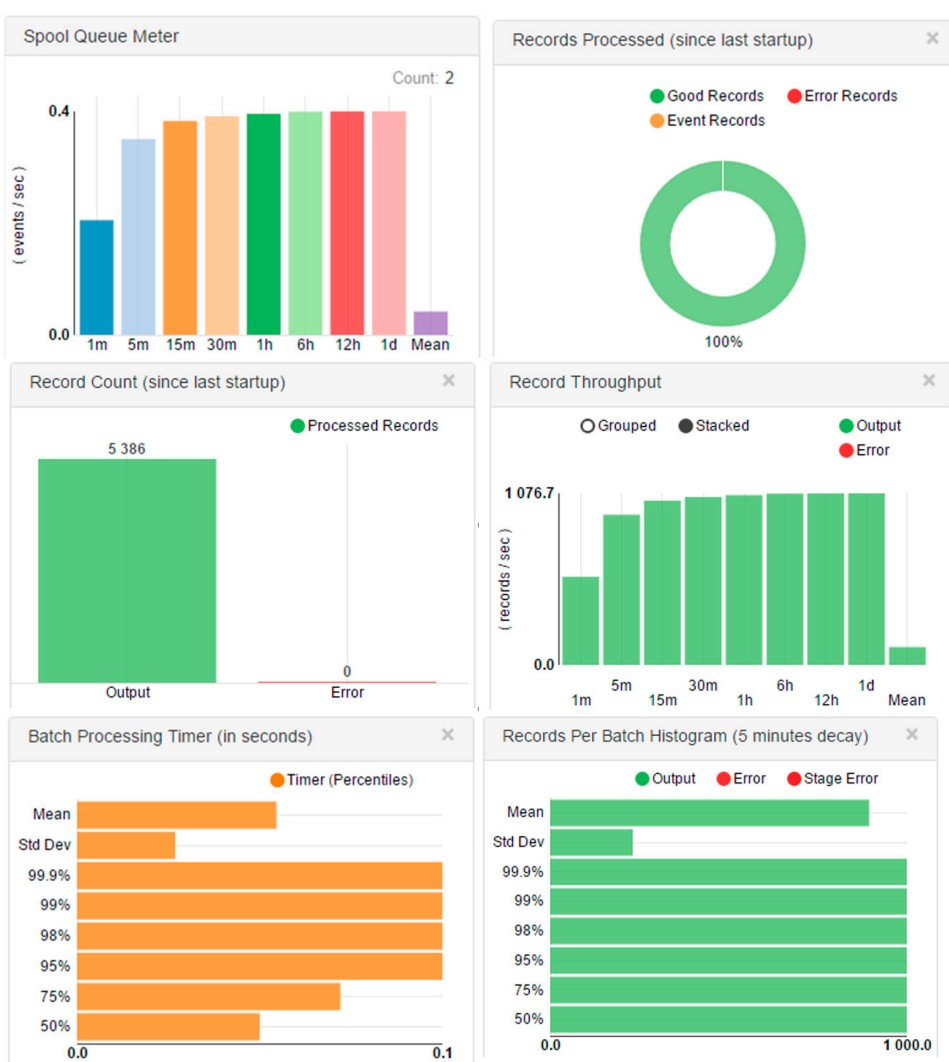

**Figure 7.** The interactive charts of the data origin outlining the number of records and intensity, along with stream/batch processing time.

The real-time database management system Apache Cassandra supports various database import and export mechanisms, allowing the querying of data from the database using external applications running either local or remote fog nodes. An example of these mechanisms is Open Database Connectivity (ODBC), providing access to databases for further analysis. The analysis in edge analytics allows other participants, either humans or machines, to make rational decisions based on the tendency witnessed in the real-time data, which allows for the monitoring and prevention of potential disasters, or other unexpected circumstances. For this reason, edge analytics, as compared to cloud analytics performed in the remote cloud, have paramount importance in the prevention of and reaction to events in the extremely early stages. Due to immediate actions, a tremendous number of natural and anthropogenic catastrophic accidents would be recognized before their proliferation, thereby saving thousands, if not millions, of people's lives. One implementation of data aggregation in the fog nodes is the gathering of pipeline real-time data from seismometers residing in seismograph stations. Their data are classified and routed to specific destinations, including a database. Owing to Apache Cassandra's support of real-time data storage, the other fog nodes querying the database for the existence of new records concerning new earthquake details can also provide a real-time data visualization (Figure 8).

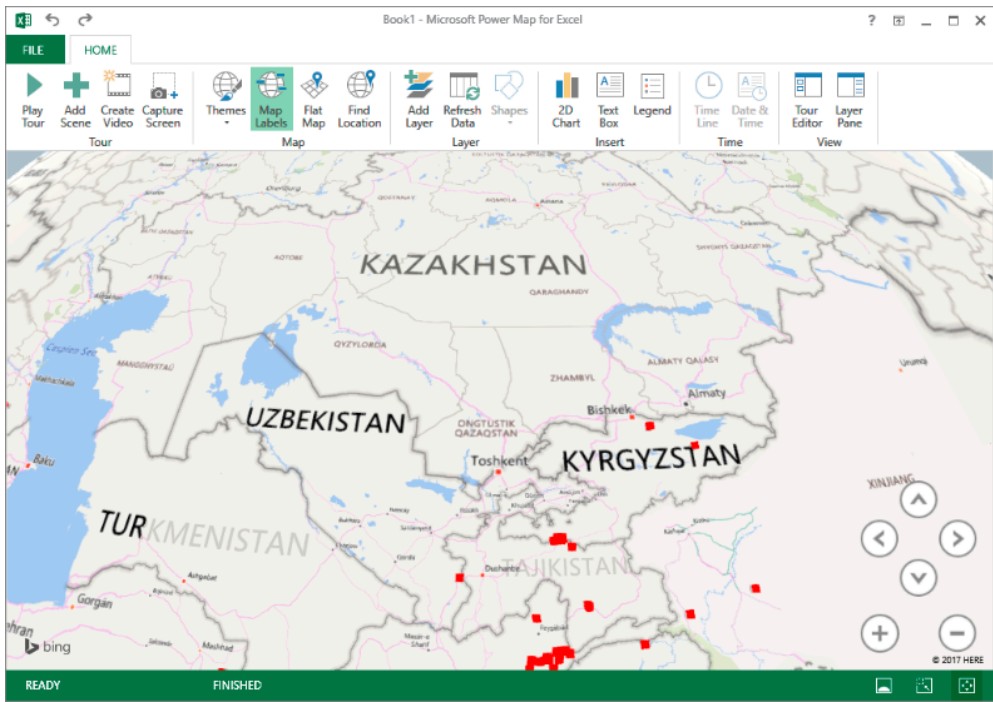

**Figure 8.** The real-time data visualization based on earthquake data from Apache Cassandra.

## 5. Analysis and Discussion of the Results

Before the post-processing of experimental data, the measure of central location recognition should be chosen. The reason for this is that, from the perspective of statistics, the mean, which is also known as the average, is not always appropriate. Therefore, another valid measure of central tendency (median) is calculated and chosen as a primary criterion for the comparison. The median is the average score for a set of data sorted by order of magnitude, and less affected by outliers and skewed data (Figure 9).

In the case of the performance assessment, pure UDP transport protocol was included to monitor the dependency of aforementioned network metrics on the upper-layer protocols used for the transmission of raw real-time data to fog nodes for further processing real-time processing and storage. The results of the experiment outline that, despite being based on User Datagram Protocol (UDP), Constrained Application Protocol (CoAP) has overtaken even its foundation UDP, and withstood the intense competition from other counterparts.

However, pure UDP without upper-layer protocols is not suggested for usage, as security and error correction mechanisms are not utilized.

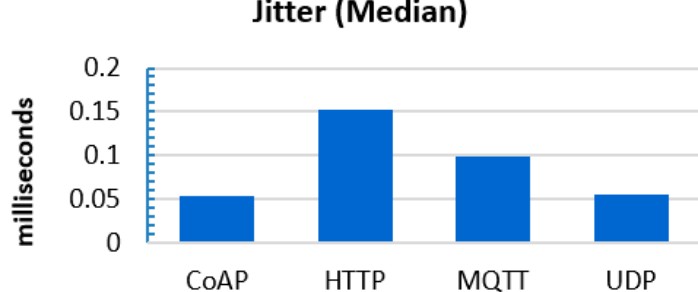

**Figure 9.** The comparison of median delay variations of four protocols suitable for big data provisioning in FC.

The use of the Transmission Control Protocol affects a higher delay variation in HTTP, at 0.153 ms, which is the highest value amongst participants, and deemed as the least appropriate protocol for big data provisioning in FC because it still follows the constraints in the value of jitter concerning real-time data. Finally, MQTT demonstrates a value of jitter at 0.099 ms, which makes it the third best possible candidate for the provision of big data (as given in Table 1).

**Table 1.** The jitters of four protocols in the experimental fog computing network.

| Protocol | Jitter (ms) | |
|---|---|---|
| | Median | Mean |
| CoAP | 0.053 | 0.07995726 |
| HTTP | 0.153 | 0.45754430 |
| MQTT | 0.099 | 0.47545580 |
| UDP | 0.055 | 0.12100310 |

A further performance evaluation is devoted to the batch processing abilities of fog nodes in fog computing, in order to scrutinize whether fog computing can tackle this task in the same way as conventional cloud computing. For this purpose, two estimations were conducted, namely, the estimation of Pi using the quasi-Monte Carlo method, and the prominent example of batch processing called 'Word Count'. The results of both experiments are provided in Tables 2 and 3.

**Table 2.** The time spent on calculating pi using the quasi-Monte Carlo method.

| Fog Node | The Number of Partitions | | | | |
|---|---|---|---|---|---|
| | 10 | 16 | 26 | 32 | 64 |
| Desktop (s) | 3.700 | 3.760 | 3.780 | 3.777 | 3.790 |
| Raspberry Pi 3 Model B (s) | 12.126 | 12.070 | 12.168 | 12.040 | 12.209 |

**Table 3.** The time spent on a batch processing of datasets of various sizes.

| File Size (Bytes) | 1,226,438 | 35,926,176 | 148,610,460 | 371,526,148 |
|---|---|---|---|---|
| | Small | Medium | Large | Bigger |
| Single node (s) | 83.007 | 504.689 | 809.096 | 2208.476 |
| Two nodes (s) | 75.084 | 116.434 | 336.209 | 1245.088 |
| Three nodes (s) | 65.120 | 84.393 | 219.528 | 568.000 |

Following the previous examination, fog nodes based on Raspberry Pi 3 Model B single-board computers were joined into a cluster running Apache Hadoop and Apache Spark.

Concerning the data above, they demonstrate that the performance of single fog nodes is limited to some extent; thus, cannot be compared with cloud computing's high performance in batch processing. This trend is arguably attributed to the heterogeneous character of fog computing involving a vast number of devices having either similar, or distinguishing, hardware characteristics.

The data in the Table 2 demonstrates that as the number of nodes in fog computing increases, the time spent on batch processing decreases exponentially. Meanwhile, the efficiency of batch processing improves proportionally to the size of the file (Figure 10).

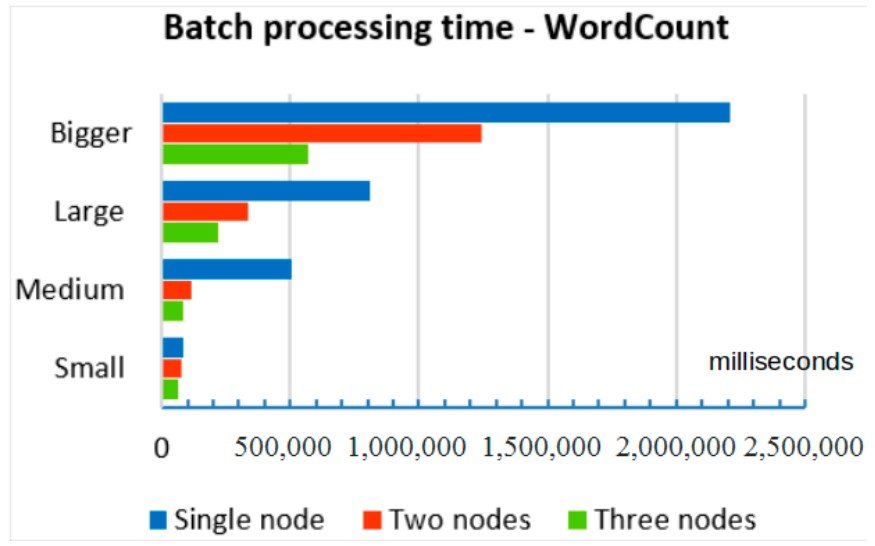

**Figure 10.** The comparison of batch processing time in four scenarios with different data sizes.

The following results given in Table 3 are derived from the experiment (measures are given in seconds):

Due to the high amount of time spent processing data at rest, it is not recommended to delegate batch processing to fog computing. The reason is that the ability of cloud computing to process this data is much faster compared to fog nodes, which have modest amounts of computing resources. However, they can generally fulfill the requirements of real-time streaming processing.

## 6. Conclusions

The paradigm of fog computing is gaining increasing popularity, with the ability to complement cloud solutions by scaling and optimizing the IoT infrastructure, being especially efficient in solving real-time problems. Fog computing is seen as an extension of the cloud at its core, which has some specific computing resources and data storage registers, and is located at the nodes of networks. With its flexible ability to process data closer to where it is produced, fog computing significantly overcomes the problem of extensive data volume, diversity, and speed.

In this paper, the capabilities of fog computing for big data provisioning were analyzed with an appropriate visualization scenario. The capabilities of fog computing for big data provisioning, such as the ability of batched stream processing, were evaluated, taking into account the above-mentioned prerequisites in terms of time using different protocols, types of nodes, and partitions; and found that the CoAP protocol is the most applicable protocol for this context in terms of jitter, while HTTP protocol shows a higher jitter time. Evaluations show that fog can be used for real-time data streaming; however, it is ineffective for batch processing.

In further works, the capabilities of fog computing in machine-learning and predictive analysis will be elaborated.

**Author Contributions:** Conceptualization, H.K. and K.A.; methodology, N.U.; software, J.K.; validation, H.K. and J.K.; project administration and supervision, M.A. (Mai Alduailij); funding acquisition, M.A. (Mona Alduailij); writing—original draft preparation, H.K.; writing—review and editing, K.A., M.A. (Mai Alduailij) and M.A. (Mona Alduailij). All authors have read and agreed to the published version of the manuscript.

**Funding:** This research received no external funding.

**Institutional Review Board Statement:** Not applicable.

**Informed Consent Statement:** Not applicable.

**Data Availability Statement:** Not applicable.

**Conflicts of Interest:** The authors declare no conflict of interest.

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
