# Peer review of "Fog Computing Capabilities for Big Data Provisioning: Visualization Scenario"

_sustainability, doi:10.3390/su14138070_

Round 1
Reviewer 1 Report
This is a very interesting research, but will be improved using a more detailed mutivariable analysis.
Is very important review the correct perspective of Hadoop as in:
Jad Darrous, Shadi Ibrahim: Understanding the performance of erasure codes in hadoop distributed file system. CHEOPS@EuroSys 2022: 24-32
Jongbaeg Lee, Jongwuk Lee, Sang-Won Lee: Optimizing Read Operations of Hadoop Distributed File System on Heterogeneous Storages. J. Inf. Sci. Eng. 37(3): 709-729 (2021)
Mohamed Elkawkagy, Heba Elbeh: High Performance Hadoop Distributed File System. Int. J. Networked Distributed Comput. 8(3): 119-123 (2020)
The interpretation of results is very importat too.
Reviewer 2 Report
- The motivation and research question is not clearly elaborated.
- It is recommended that the authors clearly present a list of their contributions through this paper.
- It is recommended that the authors present a comparison of the proposed work with the existing works.
- Latest references from good sources need to be included.
- Almost one-third self citations are noted. What are the differences of the proposed work when compared to the self cited works?
- The conclusion section is very weak and needs to be rewritten with highlight on major findings.
- The paper is weak as far as the contribution to the scientific community is concerned. It does not seem to meet the depth and level expected by the audience of this journal.
- Authors should understand and mention that not IoT is the only source of Big Data.
- Correction needed: ‘in- influential’
- The use of double quotes need to be preferably replaced with single quotes unless for cases where verbatim text is cited.
- All the figures with graphical content need to have captions for both the axes in addition to the axes labels.
- The match of the paper’s contents with the journal’s theme of sustainability needs to be elaborated.
- The paper is mostly focused on literature review with a description of short experimental results.
- It is recommended that the entire paper use the same font style and size.
- It is recommended that the tables be not cited with roman numbers, e.g. Table I, II, etc.
- All citations to the tables and figures should be with their respective numbers instead of using words like ‘above’, and ‘below’.
Reviewer 3 Report
The paper presents analysis upon fog computing usages for big data processing scenarios. The paper provides detailed introductions and experiments for FC capabilities in data analytics.
However, as an evaluation of fog computing, the paper should better frame its contributions and state why those contributions are novel and important. The author can consider to add a paragraph at the end of Section 1 for the purpose.
Also, the paper should have solid comparisons for different fog computing frameworks in terms of big data processing. The comparison can be theoretically presented and some tables can be added to showcases different metrics between each fog computing solutions. And the theoretical analysis can be complementary for the following experiments.
In addition, the paper should be carefully proof-read to fix grammar and editing issues, for example, the first sentence of second paragraph in Section 2 is in very compact space; figure 5 and figure 6 should be aligned with same boundaries as other figures, page 8 has half empty space, etc.
Round 2
Reviewer 2 Report
1. All figures with graphical display need to have the captions for both the axes, in addition to the labels of the axes.
2. Section 1 and Section 2 are in ALL UPPER CASE while remaining sections have different cases ranging from Title Case to Sentence case. Consistency is recommended.
3. It is recommended that the authors place the original figures (which are not stretched or shrinked) in the manuscript.
4. It is recommended that the authors present a summary of literature review in tabular format. Possibly the proposed work could be included as a last row of the table. This will go a long way in emphasizing the research gaps and highlighting the novel contributions through the proposed research work.
Author Response
RESPONSE TO REVIEWERS’ COMMENTS
Date: 15-06-2022
To
Editor-in-chief,
International Journal of Sustainability, MDPI
Dear Sir,
Thank you very much for reviewing our manuscript entitled “Fog Computing Capabilities for Big Data Provisioning: Visualization Scenario”. We also greatly appreciate the reviewers for their complimentary comments and suggestions. We rigorously reviewed the manuscript and removed typos and other common mistakes.
Comments and Reply:
Query #:
All figures with graphical display need to have the captions for both the axes, in addition to the labels of the axes.
Reply:
Please note that figures are taken as simulation tool output with appropriate captions according to the indication and description in the text.
Query #:
Section 1 and Section 2 are in ALL UPPER CASE while remaining sections have different cases ranging from Title Case to Sentence case. Consistency is recommended.
Reply:
Titles of sections are updated with similar cases.
Query #:
It is recommended that the authors place the original figures (which are not stretched or shrinked) in the manuscript.
Reply:
Figures in the paper mainly based on schemes and charts during the simulation fulfilled for the task statement. However, authors tried to present some figures (fig.2, fig.4 ) in better quality as given in the paper.
Query #:
It is recommended that the authors present a summary of literature review in tabular format. Possibly the proposed work could be included as a last row of the table. This will go a long way in emphasizing the research gaps and highlighting the novel contributions through the proposed research work.
Reply:
Authors made significant efforts in preparing and compiling this paper, including the review of related bibliography. Hopefully the relevance and number of literature sources are duly provided and enough for this stage and type of research given so far.
We hope that you find our responses satisfactory and that the manuscript is now acceptable for publication. We look forward to the outcome of your assessment.
With best regards,
Yours sincerely,
Halimjon Khujamatov, Khaleel Ahmad, Nargiza Usmanova, Jamshid Khoshimov, Mai Alduailij and Mona Alduaili
